# Neighbourhood prevalence-to-notification ratios for adult bacteriologically-confirmed tuberculosis reveals hotspots of underdiagnosis in Blantyre, Malawi

**McEwen Khundi**[1,2]*, **James R. Carpenter**[1], **Elizabeth L. Corbett**[1,2], **Helena R. A. Feasey**[1,2], **Rebecca Nzawa Soko**[1,2], **Marriott Nliwasa**[1,3], **Hussein Twabi**[3], **Lingstone Chiume**[1], **Rachael M. Burke**[1,2], **Katherine C. Horton**[2], **Peter J. Dodd**[4], **Ted Cohen**[5], **Peter MacPherson**[1,2,6]

1 Malawi-Liverpool-Wellcome Trust Clinical Research Programme, Blantyre, Malawi, 2 London School of Hygiene and Tropical Medicine, London, United Kingdom, 3 Helse Nord TB Initiative, College of Medicine, University of Malawi, Zomba, Malawi, 4 School of Health and Related Research, University of Sheffield, Sheffield, United Kingdom, 5 Yale School of Public Health, New Haven, CT, United States of America, 6 Department of Clinical Sciences, Liverpool School of Tropical Medicine, Liverpool, United Kingdom

* mcewenkhundi@gmail.com

**Data Availability Statement:** Data and analysis code is available at GitHub repository: https://github.com/mcewenkhundi/TBhotspotclustersplos.

## Abstract

Local information is needed to guide targeted interventions for respiratory infections such as tuberculosis (TB). Case notification rates (CNRs) are readily available, but systematically underestimate true disease burden in neighbourhoods with high diagnostic access barriers. We explored a novel approach, adjusting CNRs for under-notification (P:N ratio) using neighbourhood-level predictors of TB prevalence-to-notification ratios. We analysed data from 1) a citywide routine TB surveillance system including geolocation, confirmatory myco-bacteriology, and clinical and demographic characteristics of all registering TB patients in Blantyre, Malawi during 2015–19, and 2) an adult TB prevalence survey done in 2019. In the prevalence survey, consenting adults from randomly selected households in 72 neighbour-hoods had symptom-plus-chest X-ray screening, confirmed with sputum smear microscopy, Xpert MTB/Rif and culture. Bayesian multilevel models were used to estimate adjusted neighbourhood prevalence-to-notification ratios, based on summarised posterior draws from fitted adult bacteriologically-confirmed TB CNRs and prevalence. From 2015–19, adult bacteriologically-confirmed CNRs were 131 (479/371,834), 134 (539/415,226), 114 (519/463,707), 56 (283/517,860) and 46 (258/578,377) per 100,000 adults per annum, and 2019 bacteriologically-confirmed prevalence was 215 (29/13,490) per 100,000 adults. Lower educational achievement by household head and neighbourhood distance to TB clinic was negatively associated with CNRs. The mean neighbourhood P:N ratio was 4.49 (95% credible interval [CrI]: 0.98–11.91), consistent with underdiagnosis of TB, and was most pronounced in informal peri-urban neighbourhoods. Here we have demonstrated a method for the identification of neighbourhoods with high levels of under-diagnosis of TB without the requirement for a prevalence survey; this is important since prevalence surveys are expensive and logistically challenging. If confirmed, this approach may support more efficient and effective

**Funding:** This work was supported by two grants from Wellcome Trust (ELC grant number WT200901/Z/16/Z) and (PM grant number 200901/Z/16/Z). JRC was funded by UK Medical Research Council (MRC) programme grant MC_UU_00004/07. PJD was supported by a fellowship from the MRC (MR/P022081/1); this UK funded award was part of the EDCTP2 programme supported by the European Union. RMB was funded by Wellcome Trust (203905/16/Z). KCH was supported by the European Research Council (757699) and UK FCDO (Leaving no-one behind: transforming gendered pathways to health for TB). TC was supported by US NIH R01 R01AI147854. The funders had no role in study design, data collection and analysis, decision to publish, or preparation of the manuscript.

**Competing interests:** The authors have declared that no competing interests exist.

targeting of intensified TB and HIV case-finding interventions aiming to accelerate elimination of urban TB.

## Introduction

Despite substantial investment under global health initiatives, progress towards ending tuberculosis (TB) epidemics has been disappointingly slow. TB remained the leading infectious cause of adult death in 2019, with an estimated 1.4 million deaths, and was second only to COVID-19 in 2020 [1, 2]. Estimated incidence has been falling, but not rapidly enough to meet the EndTB Strategy goals [1, 2]. The World Health Organization (WHO) African Region achieved a 19% reduction in TB incidence between 2015 and 2020, mainly attributable to improving HIV and TB prevention and treatment services [1, 2]. Early diagnosis of TB is essential for prevention of TB deaths and new infections, as undiagnosed TB patients can remain infectious for many years if not effectively treated [3]. In 2020, however, 4.1 million (or 41% of all incident TB patients), globally, were estimated to remain undiagnosed or unnotified–with substantial increases in these "missing millions" partly because of COVID-19 disruptions in TB diagnostic services [1, 2, 4].

Reaching the ambitious WHO EndTB Strategy targets for incidence (90% reduction from 2015) and death (95% reduction from 2015) from TB by 2035 will require innovative strategies [2, 5]. Efficient diagnosis of self-presenting patients reporting TB symptoms at health facilities, although critical to patient management, is unlikely to be sufficient unless accompanied by community-based interventions [3, 6]. Active case-finding for undiagnosed TB disease (ACF), using approaches such as door-to-door enquiry for chronic cough, can rapidly reduce undiagnosed TB prevalence, but is limited to very high prevalence populations by the cost and performance of currently available TB diagnostics [3, 6–8].

Previous attempts to define "hotspots" of TB disease in urban and rural Africa and Asia have used locally-resolved case notification rates [9] that cannot distinguish poorer access to TB diagnostic services from true low disease burden [10, 11]. True TB burden is highly heterogeneous but, even in the same District or City, substantial heterogeneity in routine TB service access [3, 12] tends to obscure hotspots, which typically represent the combined effects of adverse social and environmental determinants with high barriers to accessing health services [3, 13, 14].

Data from TB prevalence surveys can provide detailed neighbourhood-level data on TB determinants and undiagnosed TB burden to guide National TB Programmes and District Health Officers [9, 12, 15, 16], allowing neighbourhoods with delayed detection and incomplete detection [16, 17] of TB to be identified through high prevalence-to-notification (P:N) ratios [17, 18]. TB prevalence surveys are, however, costly and logistically demanding. Identification of neighbourhoods with high levels of under-diagnosis without requirement for prevalence surveys could be of major benefit to National TB Programmes. Here, we aimed to develop simple, accurate models that could be used by researchers and TB Programme Managers using high-quality spatially-resolved TB neighbourhood level data from urban Blantyre, Malawi. We used multilevel Bayesian modelling to generate neighbourhood level P:N ratios [9, 17], aiming to smooth over sparse prevalence data and borrow strength across neighbourhoods and make predictions beyond available prevalence data to support better prioritisation of community-based ACF.

## Methods

### Study setting

Blantyre District is in the Southern Region of Malawi, Central Africa. Blantyre City, in the centre of the District, is a major commercial centre and has several densely-populated informal settlements, as well as more-established urban and peri-urban neighbourhoods where rates of poverty are high and access to municipal and health services are limited [19]. In the 2018 Malawi National census the Blantyre City population was 800,264 (502,018 adults ≥ 15y) [20] and adult HIV prevalence was 18% in a recent population-based survey [21].

### Blantyre enhanced TB monitoring and evaluation of TB notifications

Patients diagnosed with TB in Malawi are registered by the National TB Programme. TB registration clinics in Blantyre include a government referral hospital, free public clinics and a small number of private health facilities [22]. TB Officers (a cadre of health workers employed by Malawi's Ministry of Health with responsibility for delivering TB services) [22] were supported to strengthen the TB notification surveillance system in Blantyre as part of a joint project between the Malawi-Liverpool-Wellcome Trust Clinical Research Programme (MLW), the Blantyre District Health Office, and the Malawi National TB Control Programme (NTP). All TB patients are offered provider-initiated HIV testing and antiretroviral therapy if newly diagnosed with HIV [22].

From 2015, TB Officers have been supported to use an electronic data capture application (ePaL) to collect additional clinical, sociodemographic, and household level data and a confirmatory sputum for microscopy and culture as part of the citywide enhanced surveillance system [4, 11]. ePAL supports capture of global positioning satellites (GPS) coordinates identifying the place of residence for each TB patient, using high resolution satellite maps with locally-captured reference locations within each neighbourhood of the city. The ePAL application has previously been validated and described elsewhere [11, 23]. All patients starting TB treatment were asked to provide an additional single spot sputum sample for smear microscopy and mycobacteria growth indicator tube (MGIT) culture, performed at the MLW/University of Malawi College of Medicine TB Research Laboratory. The enhanced TB surveillance data and NTP registers were reconciled on a quarterly basis, and monthly 5% of patients were traced to home for data validation purposes.

### TB prevalence survey

In 2019, a TB prevalence survey was carried out in Blantyre City by the MLW study team at the start of a planned cluster-randomised trial of community-based TB screening interventions, subsequently interrupted by COVID-19 (ISRCTN11400592). 72 neighbourhood clusters were defined, each comprised of several community health workers (CHW) areas, with the goal of having approximately 4000 adults in each neighbourhood. CHW areas are the smallest health administrative unit in the city, and each is affiliated to a primary health clinic [11, 24]. Using Google Earth, a geographical information specialist captured the GPS coordinates of all the houses in the 72 neighbourhood clusters to be used as the prevalence survey sampling frame. In each neighbourhood, 115 households were selected at random for participation into the prevalence survey with the aim of recruiting 215 adults (≥18 years old) per neighbourhood.

Adults from the randomly selected households were visited and invited to attend a study tent located at a central point within the neighbourhood for TB and HIV investigations. TB screening was provided at the tent using a digital chest radiograph that was immediately read

by an experienced radiographer trained in TB prevalence surveys, and with interpretation supported by computer assisted diagnostic software (Qure.ai version 2.0). Participants who had an abnormal chest X-ray or reported cough of any duration were asked to provide two spot sputum samples for Xpert MTB/Rif, smear microscopy, and MGIT culture. Positive Xpert MTB/Rif and smear microscopy results were provided within two days, and culture results within approximately six weeks; a prevalent TB case was defined as a positive result for Xpert MTB/Rif or smear microscopy, with a positive MGIT culture result that was speciated as *Mycobacterium tuberculosis*. HIV testing was offered using both OraQuick (OraSure Technologies, manufactured in Thailand) oral HIV test kits and a rapid fingerprick kits (Determine 1/2, Alere, USA) in parallel. Positive HIV results were confirmed using Uni-Gold (Trinity Biotech, Ireland). If participants verbally reported being HIV positive, only a Uni-Gold confirmation test was done. All participants with newly diagnosed HIV were provided with post-test counselling and assisted to register for HIV treatment at their nearest primary care clinic.

## Neighbourhood populations

In 2015, the MLW study team conducted a population census in all of the city's CHW areas [11]; in an independent exercise the Malawi National Statistical Office (NSO) conducted the Malawi Population and Household National Census in 2018 that included Blantyre City [20]. Population denominators were derived from the MLW study team's 2015 census and the NSO's 2018 census. The 2018 NSO and 2015 MLW study census data were used to calculate annual population growth rates, which were then used to estimate annual population denominators for the 72 study neighbourhoods from 2015 to 2019. The growth rates were calculated separately for ages 0 to 4yrs, 5 to 14yrs, 15yrs or older adult males and 15yrs and older adult females. Growth rates were assumed to be the same for all the neighbourhoods.

## Calculation of empirical TB notification and prevalence rates and predictors

Empirical neighbourhood-level TB case notification rates (CNR) from 2015 to 2019 and 2019 TB prevalence rates were calculated by summing the adult ($\geq$18y) TB notifications or prevalent cases at neighbourhood level and dividing by respective adult population denominators and multiplying by 100,000 to scale the rate to per 100,000 population. The percentage of adults in each neighbourhood and the percentage of male adults aged 15 years or older were calculated using data from the 2015 study team Blantyre City census and were assumed to be consistent from 2016 to 2019. The distance to the nearest TB clinic was estimated by calculating the cartesian (straight line) distance between the centroid of each neighbourhood and the nearest TB clinic; this served as a proxy indicator for access to TB diagnosis and treatment [25]. Neighbourhood HIV prevalence was calculated using prevalence survey data, and the percentage of households whose head never finished primary school—a proxy variable for poverty—was estimated using data from the TB prevalence study. According to the Malawi Integrated Household Survey, household head education level was closely associated with household poverty [26, 27].

## Neighbourhood baseline characteristics

We report the neighbourhood-level percentage of adults aged 15 years or older, percentage of male adults (15 years or older), distance to nearest TB clinic, percentage of households with head of house who never finished primary school and HIV prevalence, summarised by their mean, range, and standard deviation (sd). We plotted the spatial distribution of the covariates across the 72 neighbourhood clusters on choropleth maps.

## Statistical modelling

We fitted Bayesian multilevel models to estimate neighbourhood-level adult ($\geq$ 18y) annual bacteriologically-confirmed TB case notification rates for each year between 2015 and 2019, and separately for prevalence rates for 2019. Models were fitted using Markov chain Monte Carlo (MCMC) sampling using the brms package as an interface to Stan in R, with inference based on three chains of 14,000 posterior samples after discarding 1000 burn-in samples [28]. Since the notification and prevalence data were from two different and independent datasets they were modelled separately to allow greater control in exploring covariates. The bacterio-logically-confirmed case notification data were modelled using a Poisson response distribution (Eq 1), we included a dummy variable for year with the reference level of 2019.The prevalence data were modelled using a zero-inflated-Poisson distribution (Eq 2) to account for overdispersion and excess neighbourhoods with zero prevalent cases. Neighbourhood-level random effects were modelled with a spatial intrinsic conditional autoregressive (ICAR) term (S1 and S2 Equations) or a random intercept term (Eqs 1 and 2), but not with both random terms in the same model. Possible combination of neighbourhood-level variables were derived for the TB case notification and prevalence data. The set of variables considered for inclusion in the model were selected based on previous research [11], ease of measurement for TB pro-grammes, and availability within the datasets. The models' predictive performance were evaluated using leave-one-out (LOO) cross-validation [29]. The best fitting models were selected based on their expected log pointwise predictive density (ELPD) LOO statistic [29] (S5 and S6 Tables). Weakly regularising priors were assigned to model intercepts and slopes. Model convergence was assessed by visual inspection of trace plots, effective sample sizes and Gelman-Rubin statistics [28].

**Equation 1**

Let $Y \sim Pois(\mu)$

$$\Pr\left(Y_{ij} = y_{ij}\right) = \left(\frac{\mu_{ij}^{y_{ij}} \, exp(-\mu_{ij})}{y_{ij}!}\right)$$

$$\log(\mu_{ij}) = \alpha + \alpha_i + \beta_1 x_{1i} + \beta_2 x_{2i} \ldots + \beta_k x_{ki} + \beta_{year2015} year2015_j + \beta_{year2016} year2016_j$$
$$+ \beta_{year2017} year2017_j + \beta_{year2018} year2018_j + \log(Pop_{ij})$$

$$\alpha \sim Normal(\mu_\alpha = 0, \sigma_\alpha^2 = 10)$$

$$\beta_k \sim Normal(\mu_\beta = 0, \sigma_\beta^2 = 10)$$

$$\alpha_i \sim Normal(0, \sigma_{\alpha_i}^2)$$

$$\sigma_{\alpha_i} \sim HalfCauchy(0, 1)$$

The expectation of $Y_{ij} = y_{ij}$ given by:

$$E(Y_{ij} = y_{ij}) = \mu_{ij}$$

Where i refers to neighbourhood i for i = 1,2,3..72, and $j$ indexes year (2015,...2019), ($\alpha$ +$\alpha_i$) intercept parameters and $\beta_1, \beta_2 \ldots \beta_k$ are unknown regression coefficients that are estimated from the data for the cluster level covariates $x_{1i}, x_{2i} \ldots x_{ki}$, and sigma ($\sigma_{\alpha_i}$) is the standard deviation for the random intercept for neighbourhood i. $Pop_{ij}$ is the total population of

neighbourhood in year $j$ that is used as the offset. Here $year2015_j$, $year2016_j$, $year2017_j$ and $year2018_j$ are dummy variables which take the value of one for that year and take the value of zero for the other years (the baseline year was 2019).

**Equation 2**

To define the zero-inflated Poisson model, let $Z{\sim}Bern(1{-}p)$ (so $Pr\{Z = 0\} = p$) and independently $W{\sim}Pois(\mu)$. Then the data are modelled by the zero-inflated variable Y which is defined as

$$Y = ZW$$

$$\Pr(Y_i = y_i) = \begin{cases} (p + (1-p)exp(-\mu_i)) \; if \; y_i = 0 \\ \left((1-p)\frac{\mu_i^{y_i}exp(-\mu_i)}{y_i!}\right) \; if \; y_i \geq 1 \end{cases}$$

$$logit(p_i) = \vartheta$$

$$\vartheta \sim Normal(\mu = 0, \sigma_\vartheta^2 = 10)$$

$$log(\mu_i) = \alpha + \alpha_i + \beta_1 x_{1i} + \beta_2 x_{2i} \ldots + \beta_k x_{ki} + log(Pop_i)$$

$$\alpha \sim Normal(\mu_\alpha = 0, \sigma_\alpha^2 = 10)$$

$$\beta_k \sim Normal(\mu_\beta = 0, \sigma_\beta^2 = 10)$$

$$\alpha_i \sim Normal(0, \sigma_{\alpha_i}^2)$$

$$\sigma_{\alpha_i} \sim HalfCauchy(0, 1)$$

The expectation of $Y_i = y_i$ is given by:

$$E(Y_i = y_i) = (1-p)\,\mu_i$$

Where i refers to neighbourhood i for i = 1,2,3...72 (note we only have prevalence data for one year, 2019). $\vartheta$, $(\alpha{+}\alpha_i)$ intercept parameters and $\beta_1, \beta_2 \ldots \beta_k$ are unknown regression coefficients that are estimated from the data for the covariates $x_{1i}, x_{2i} \ldots x_{ki}$, and sigma $(\sigma_{\alpha_i})$ is the standard deviation for the random intercept for neighbourhood i. $Pop_i$ is the total population of neighbourhood that is used as the offset.

The intercept, mean rate ratios and 95% credible intervals (CrI) of the selected models are presented in a model summary table. We drew 42,000 posterior samples of the prevalence and notification rates from the selected models. The posterior bacteriologically-confirmed P:N ratios were calculated by dividing the posterior prevalence rates by the posterior notification rates (Eq 3). We obtained the posterior mean of the P:N ratios and their 95% posterior CrI, and summarised their distribution in a choropleth map and a caterpillar plot [30]. Analysis was conducted using R version 4.0.3 (R Foundation for Statistical Computing, Vienna).

**Equation 3**

$$\text{Posterior notification rate}_{ij} = \exp(\alpha_j + \alpha_{ij}) * 100,000 \text{ (From Equation 1)}$$

$$\text{Posterior prevalance rate}_{ij} = (1 - p_j) * \exp(\alpha_j + \alpha_{ij}) * 100,000 \text{ (From Equation 2)}$$

Where j indexes the posterior sample j = 1,2,3, . . . 42,000 and i is for neighbourhood i = 1,2,3 . . . 72.

The P:N ratio posterior was calculated by first drawing a sample of 42,000 posterior samples of the posterior notification and prevalence rates per 100,000. The j[th] posterior P:N ratio was calculated by dividing the j[th] posterior prevalence rate by the j[th] posterior notification rate.

**Sensitivity analysis.** Neighbourhood-level TB prevalence was post-stratified according to age-sex groups in neighbourhood populations from WorldPop in order to correct for under-participation of some age-sex groups in the prevalence survey [31]. WorldPop population estimates were used because they had more granular age-sex groups than our census-based denominators. The post stratified TB prevalence was used to reproduce P:N ratios. We additionally undertook sensitivity analysis to estimate neighbourhood P:N ratio for all adult forms of TB, including both bacteriologically-confirmed and clinically-diagnosed cases.

## Ethical considerations

Ethical approval was granted by the London School of Hygiene and Tropical Medicine (16228) and the College of Medicine, University of Malawi Research Ethics Committee (P.12/18/2556). Participants in both the prevalence survey and MLW study census provided written informed consent.

## Results

### Neighbourhood characteristics

From 2015 to 2019, the estimated total population of the 72 study neighbourhoods increased from 612,792 to 905,419, with 60.90% (range: 54.80–70.60, sd: 3.00) adults ($\geq$15 years). There was substantial neighbourhood-level variability in the mean percentage of adults who were men (51.54%, range: 46.89–55.23, sd: 1.55), living in households headed by someone who had not completed primary education (low education: 16.90%, range 4.30–32.40%, sd: 6.10) and HIV prevalence 13.80% (range: 4.21–27.44%, sd: 4.32). The mean distance from the centroid of the neighbourhoods to the nearest TB clinic was (1.74 km, range: 0.36–3.68 km, sd: 0.89) Table 1. Neighbourhoods located near the centre of the city had higher population density, and higher proportions of adults and male residents (Fig 1). The more peripheral and peri-urban neighbourhoods tended to have higher distance to the nearest TB clinic, HIV prevalence and percentage of household heads without primary education were generally higher in the more peripheral (peri-urban) neighbourhood.

### Empirical neighbourhood adult bacteriologically-confirmed TB cases notification and prevalence rates

During 2015–19 there were a total of 2,078 adults aged 18 years or older with bacteriologically-confirmed TB registered for treatment from the study neighbourhoods. CNR's declined during this period; neighbourhood mean adult bacteriologically-confirmed TB CNR's from 2015 to 2019 were 131 (range: 0–383, sd: 77), 134 (range: 32–328, sd:70), 114 (range: 28–291, sd:57), 56 (range: 0–167, sd:38) and 46 (0–144, sd:29) per 1000,000 adults Table 1. HIV prevalence

**Table 1. Neighbourhood-level summary data for the 72 neighbourhoods.**

| Characteristic | Mean (sd) | Range | n | N |
|---|---|---|---|---|
| 2015 bacteriologically-confirmed adult TB notification rate (per 100,000) | 131 (77) | 0–383 | 479 | 371,834 |
| 2016 bacteriologically-confirmed adult TB notification rate (per 100,000) | 134 (70) | 32–328 | 539 | 415,226 |
| 2017 bacteriologically-confirmed adult TB notification rate (per 100,000) | 114 (57) | 28–291 | 519 | 463,707 |
| 2018 bacteriologically-confirmed adult TB notification rate (per 100,000) | 56 (38) | 0–167 | 283 | 517,860 |
| 2019 bacteriologically-confirmed adult TB notification rate (per 100,000) | 46 (29) | 0–144 | 258 | 578,377 |
| 2019 adult bacteriologically-confirmed TB prevalence rate (per 100,000) | 215 (335) | 0–1,415 | 29 | 13,490 |
| 2015 adult TB notification rate† (per 100,000) | 242 (119) | 23–639 | 884 | 371,834 |
| 2016 adult TB notification rate† (per 100,000) | 273 (128) | 47–629 | 1,106 | 415,226 |
| 2017 adult TB notification rate† (per 100,000) | 251 (101) | 42–451 | 1,157 | 463,707 |
| 2018 adult TB notification rate† (per 100,000) | 127 (66) | 13–287 | 642 | 517,860 |
| 2019 adult TB notification rate† (per 100,000) | 150 (65) | 28–349 | 849 | 578,377 |
| Percentage of adults (≥15y) (%) | 60.90 (3.00) | 54.80–70.60 | 371,834 | 612,792 |
| Percentage of male adults (%) | 51.54 (1.55) | 46.89–55.23 | 191,855 | 371,834 |
| Household head without primary education (%) | 16.90 (6.10) | 4.30–32.40 | 2,700 | 15,897 |
| Distance to TB clinic (km) | 1.74 (0.89) | 0.36–3.68 | NA | NA |
| HIV prevalence (%) | 13.80 (4.32) | 4.21–27.44 | 1631 | 11705 |

km, kilometre; n, numerator; N, denominator; NA, (not applicable); range, minimum—maximum; sd, standard deviation. Numerator and denominators for TB notifications and TB prevalence limited to adults.

†All forms of TB

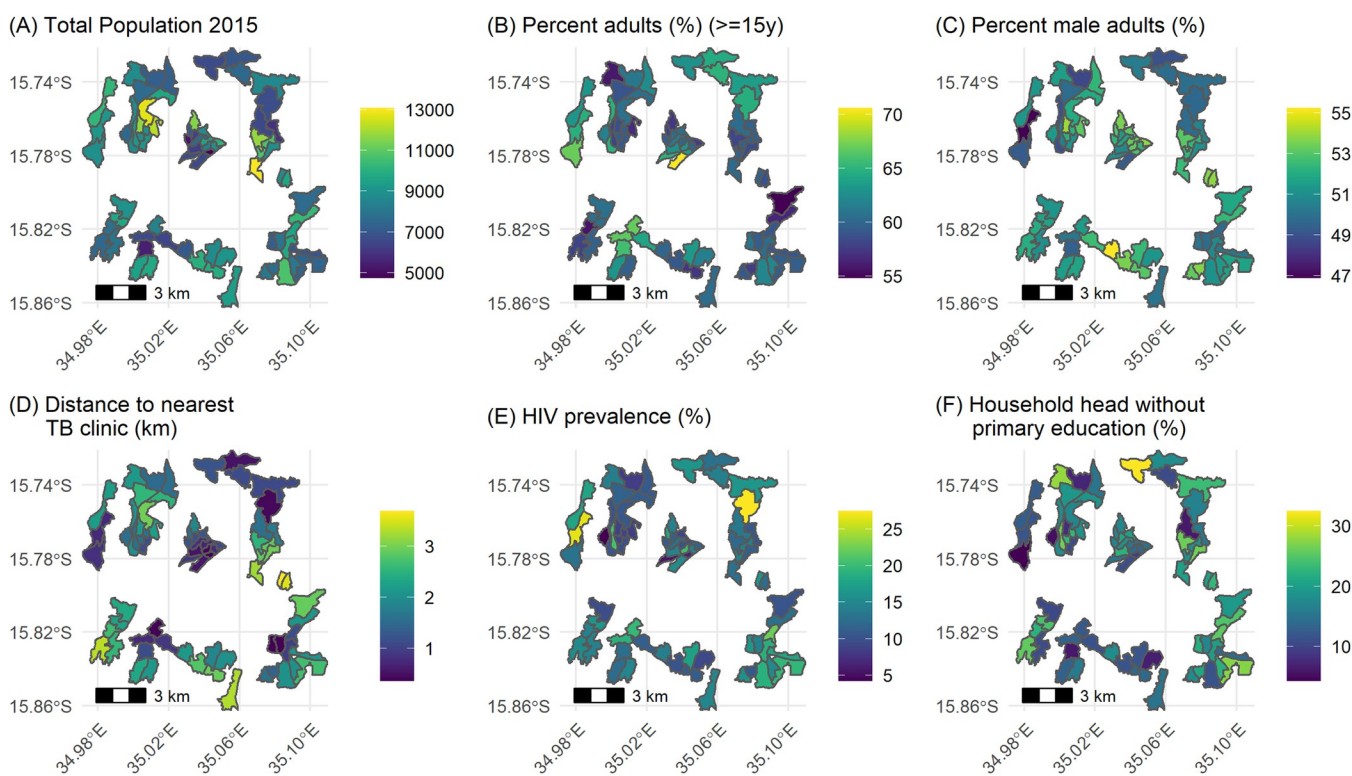

**Fig 1. Choropleth maps all covariates considered in predictive models of TB case prevalence and notification rates.**

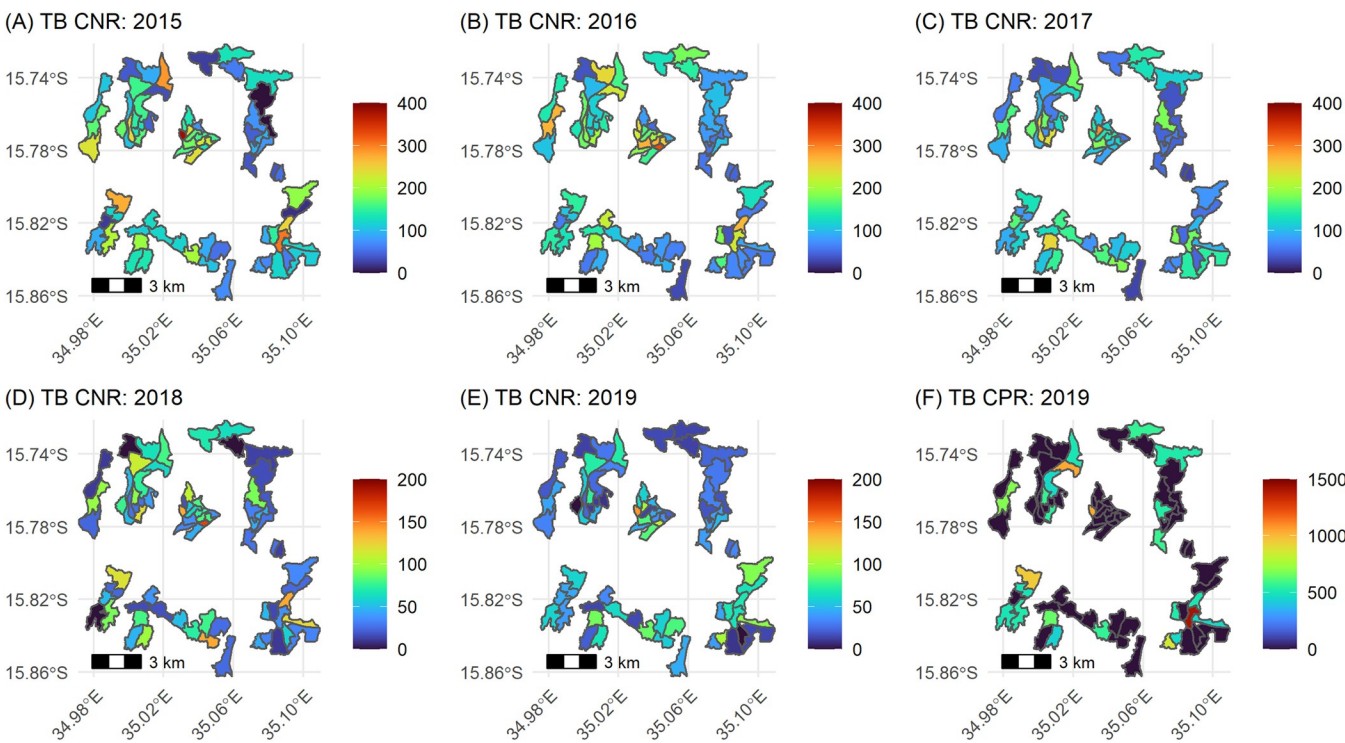

**Fig 2. Empirical bacteriologically-confirmed adult TB case notification rates (CNR) 2015–2019 (A-E), and TB case prevalence rates (CPR) 2019 (F); both per 100,000.**

among TB cases was 65.45% (3394/5260) and 64.56% (1864/5260) of registered patients were men.

A total of 29 (range: 0–3, sd: 0.6) bacteriologically-confirmed previously undiagnosed adult TB patients were identified during the prevalence survey, with an empirical neighbourhood mean of bacteriologically-confirmed TB prevalence rate of 215 (29/13,490) per 100,000 (range: 0–1,415, sd: 335). The rates of empirical TB notifications were higher in the city centre and were lower in the city's outskirts, whereas empirical TB prevalence rates were higher on the outskirts of the city Fig 2.

## Neighbourhood-level predicted TB prevalence and notifications rates

For analysis of microbiologically-confirmed adult TB CNRs, the model that was selected included a neighbourhood-level random intercept, and neighbourhood-level covariates including the percentage of adults ($\geq$15y), distance to the nearest TB clinic, and percentage of household heads who had not completed primary school education (S2 and S6 Tables).

There was an overall trend of a reduction in rates of annual adult bacteriologically-confirmed TB notification rates. In comparison to 2019, the years 2015 (rate ratio [RR]: 2.89, 95% CrI: 2.48–3.37), 2016 (RR: 2.91, 95% CrI: 2.51–3.38), 2017 (RR: 2.51, 95% CrI: 2.16–2.92) and 2018 (RR: 1.23, 95% CrI: 1.03–1.45) had substantially higher CNRs (Table 2).

For the prevalence rates, the model with a random intercept for neighbourhood and neighbourhood percentage of adults ($\geq$15y), was selected (S1 and S5 Tables). In this model, there was no association between the neighbourhood TB prevalence rate and the percentage of neighbourhood adults ($\geq$15y) (Table 2).

**Table 2. Parameter estimates for selected regression models for predicting neighbourhood level TB prevalence and notifications.**

| Fixed effects<br>Parameters | Adult bacteriologically-confirmed TB notification model | | Adult bacteriologically -confirmed TB prevalence model | |
|---|---|---|---|---|
| | Mean rate ratio | 95% CrI | Mean rate ratio | 95% CrI |
| Percentage of adult residents ($\geq$15y)[a] | 0.96 | (0.93, 1.00) | 0.94 | (0.80, 1.10) |
| Distance to nearest TB clinic (km)[a] | 0.78 | (0.69, 0.88) | | |
| Percentage of household heads that did not complete primary school[a] | 0.98 | (0.96, 0.99) | | |
| Year: 2019 | Reference | | | |
| Year: 2015 | 2.89 | (2.48, 3.37) | | |
| Year: 2016 | 2.91 | (2.51, 3.38) | | |
| Year: 2017 | 2.51 | (2.16, 2.92) | | |
| Year: 2018 | 1.23 | (1.03, 1.45) | | |
| Intercept | $50.88^*10^{-5}$ | $(42.99^*10^{-5}, 60.00^*10^{-5})$ | $232.08^*10^{-5}$ | $(132.05^*10^{-5}, 404.96^*10^{-5})$ |
| Zero inflation intercept | | | 0.18 | (0.01, 0.46) |
| *Random effects SD: cluster* | 0.31 | (0.24, 0.39) | 0.33 | (0.01, 0.90) |

CrI, Credible interval; Km, kilometre; sd, standard deviation.

[a]Percentage of adults was centred by subtracting by its mean (60.90%), Distance to nearest TB clinic (km) was centred by subtracting by 1km, Percentage of household head that did not complete primary school was centred by subtracting by its mean (16.90%).

## Neighbourhood level hotspots of TB underdiagnosis, ratio of prevalence to notifications

The mean neighbourhood posterior P:N ratios for adult bacteriologically-confirmed TB varied considerably between neighbourhoods (range: 1.70–10.40, sd: 1.79), with the mean posterior P:N ratio of the 72 neighbourhoods being 4.49 (95% CrI: 0.98–11.91) Fig 3.

Overall, P:N ratios were higher in neighbourhoods in the outskirts of the city, characterised by rapidly-growing informal settlements (Fig 4).

**Sensitivity analysis.** The model coefficients were mostly similar to that in the primary analysis (S7 and S8 Tables). The mean posterior P:N ratio of the 72 neighbourhoods was 5.04 (95% CrI: 1.86–10.26) and was 1.39 (95% CrI: 0.30–3.58) for the post stratified TB prevalence base analysis and the analysis based on all notified TB cases respectively. Overall, the distribution of the P:N ratios was similar to the primary analysis (S1–S5 Figs). The sensitivity analysis of all TB case notifications classified 14 neighbourhoods into the 4th quartile of the 18 neighbourhoods that were classified as having 4th quartile P:N ratios by the primary analysis, while the analysis based on post-stratified TB prevalence classified 16 neighbourhoods into 4th quartile of the 18 neighbourhoods that were classified as having 4th quartile P:N ratio by the primary analysis (S9 Table). Similar to the primary analysis, the P:N ratios were higher in the outskirts of the city in the informal urban settlements that were located in the rapidly-growing informal settlements on the periphery of the city (S3 and S5 Figs).

## Discussion

Our main finding was that urban Blantyre in Malawi, a city with free health services and high coverage of treatment for HIV but high rates of poverty, has a significant burden of delayed and undiagnosed TB, with an overall estimated mean neighbourhood TB P:N ratio of 4.49:1. The 18/72 neighbourhoods with P:N ratios in the highest quartile were likely to be "hotspots" of delayed diagnosis or missed diagnosis of TB cases. Most neighbourhoods with high P:N ratios were located in the rapidly-growing informal settlements on the periphery of the city and were adjacent to forest reserves or mountainous terrain where the city is expanding.

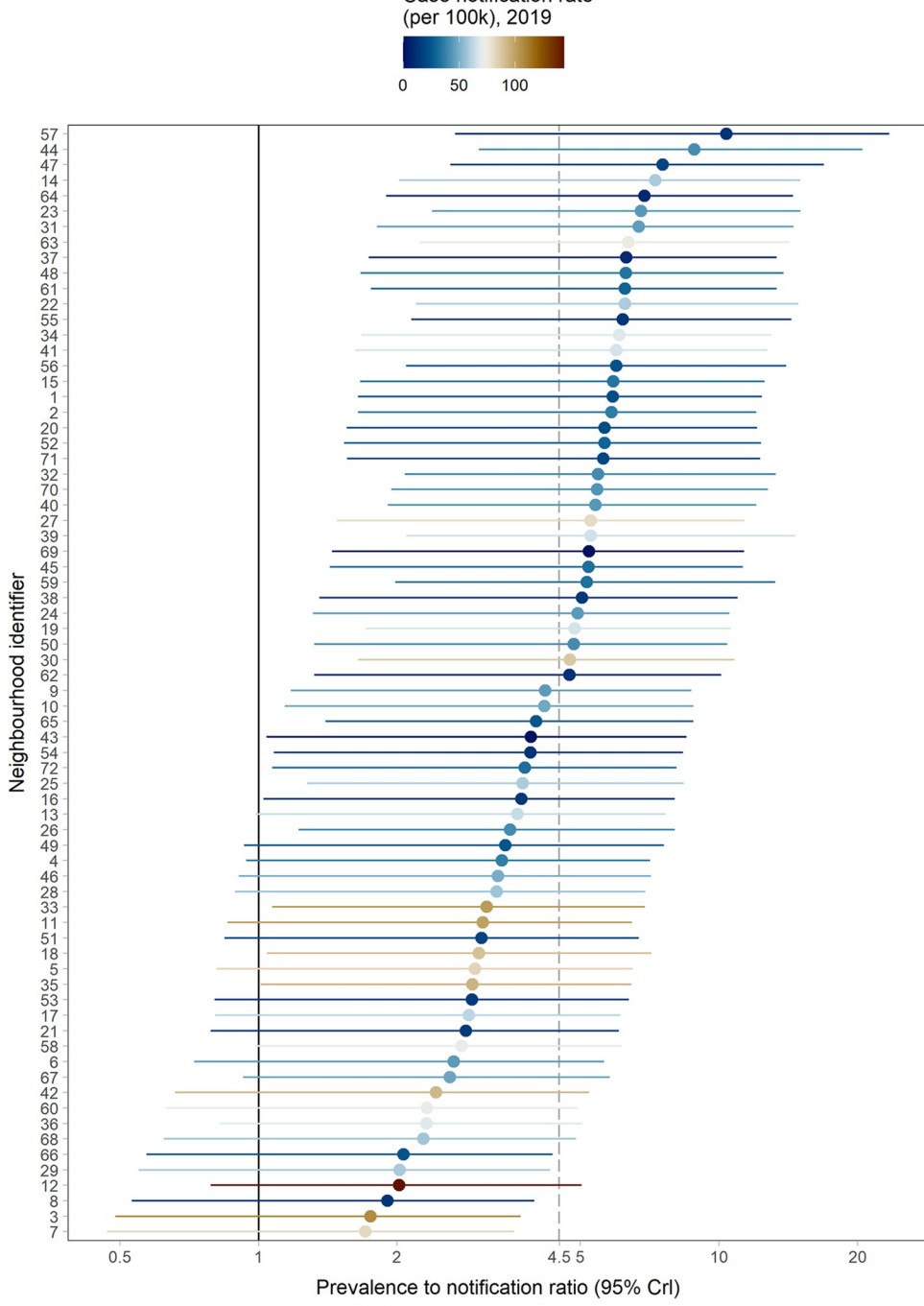

**Fig 3. Neighbourhood level TB prevalence to notification ratios (with 95% CrIs) using final models.** The neighbourhoods were ordered according to prevalence to notification ratio size. The dashed line is the mean prevalence to notification ratio. CrI Credible interval.

However, we did not directly capture informality, and further research is required to quantitatively define "the degree of formality of settlement"; these are residential areas that are not registered by the authorities or have makeshift housing structures and are likely to be important areas of focus for TB case finding activities. Our approach, which uses data that can be collected by District Health Officers in urban African cities (neighbourhood distance to clinic,

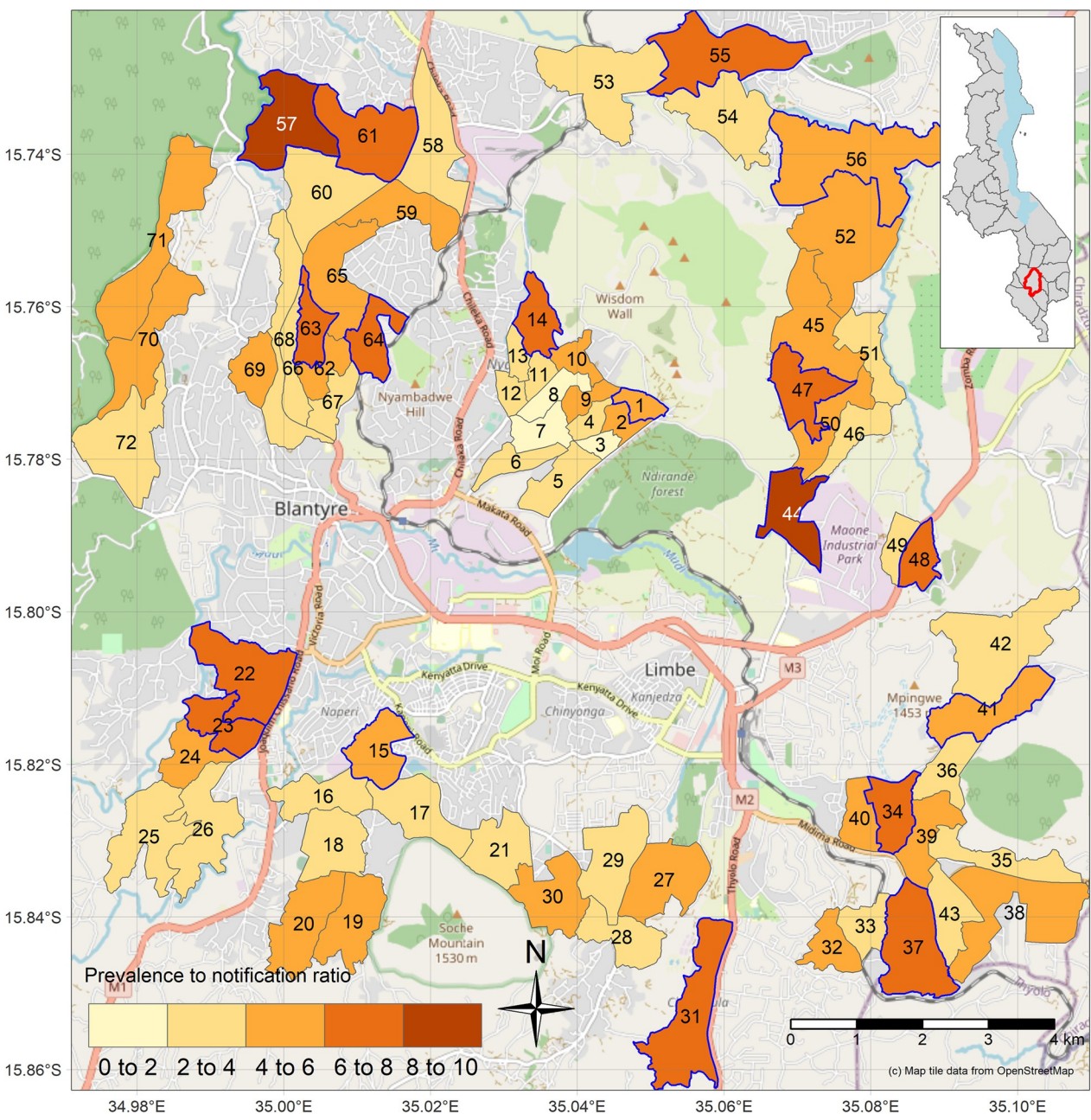

**Fig 4. Map of TB prevalence to notification ratios predicted from final models including estimated neighbourhood random effects (inset map of Malawi with Blantyre in red).** Mosdels include neighbourhood random effects. Neighbourhoods outlined in blue are in the highest quartile for P: N ratios. Inset map of Malawi with Blantyre District in red. Map tile data from OpenStreetMap.

percentage of male residents, and percentage of households where the head has not completed primary education), could be used to prioritise neighbourhoods for community-based TB ACF and prevention interventions, potentially a more efficient and effective way of delivering TB screening and prevention interventions [32].

A major strength of this analysis was that we used data from an enhanced TB surveillance system and a well-conducted subdistrict area prevalence survey. We used a systematic model-ling strategy to identify a parsimonious model with variables that are predictive of

neighbourhood TB notification and prevalence rates. Our strategy of combining notification and prevalence data offers important improvements in identifying high-burden areas compared to approaches based solely on notification rates [11, 12]. When only notifications are used, we might erroneously identify areas with easier access to TB diagnosis as hotspots and miss areas with a high burden of undiagnosed TB that have more limited access to TB diagnosis [17]. By using both notification and prevalence we can identify areas where the underlying notification of TB cases misses a higher proportion of undiagnosed TB cases [17]. But, by using the covariates identified in our models, our hope is that a new setting with similar characteristics to Blantyre might successfully rank its neighbourhoods to identify areas likely to have underdiagnosis. Hence our approach may reduce the need to carrying out a relatively expensive across-the-board prevalence survey [33].

The End TB goal of 90 percent reduction in TB incidence between 2015 and 2035 is difficult to track in the majority of high-TB settings where TB incidence cannot be estimated directly from notifications but is estimated through inference methods that can produce imprecise estimates [1, 2, 34]. Developing methods for identifying areas of TB underdiagnosis is therefore critical to guide targeted intervention [12, 35, 36], which will be more important as epidemics become more concentrated. CNRs in Blantyre Malawi declined between 2015 to 2019, reflecting the general trend in other African countries in this period [1, 2]. Blantyre has also achieved high coverage of HIV treatment and isoniazid preventive therapy [21, 37].

The P:N ratio, which is used to assess TB burden in this study, is a proxy for the time between onset of TB infectiousness and diagnosis, and it is typically given the unit of years in mathematical modelling studies; its inverse is known as the patient diagnostic rate [17, 38]. While the P:N ratio does not directly measure incidence, it can be used as an indicator of neighbourhood delayed TB case diagnoses or missed diagnosis [17]. We found overall across Blantyre that the P:N ratio was high (4.49:1), indicating substantial underdiagnosis and delayed diagnosis of TB.

Our data suggests that the efficiency of selecting communities for community-based interventions, such as those based on poverty and population density, could be improved by incorporating neighbourhood-level prevalence survey and notification data which allows a more granular understanding of TB epidemiology [3, 9, 32, 39]. District TB programmes benefit the most when they have information that enable them to prioritise their efforts, because they usually operate under resource constraints [3, 35. 39]. TB programmes must collect additional data to what is routinely collected through national TB case notification systems at the time of registering patients to gain a better understanding of the local TB epidemiology and guide public health interventions [15, 32, 36]. The majority of people who are disproportionately affected by TB, live in informal urban settlements that lack postal or zip codes [1, 11, 19]. Collecting of neighbourhood location of TB patients' households may provide additional epidemiological insights for planning spatially-targeted interventions of TB hotspots [11–13, 35, 36].

Community-based TB ACF interventions have been shown to be most effective when conducted intensively, through repeated screening of communities over a short period of time [7, 8]. For example, the ACT3 trial in Vietnam, conducted between 2014 and 2016, demonstrated reductions in TB prevalence through community ACF by offering annual Xpert MTB/Rif screening in 60 intervention neighbourhoods with approximately 54,000 adults over three years [8]. In a post intervention prevalence survey the intervention clusters showed a 44% reduction in TB prevalence [8]. TB screening interventions such as those tested in the ACT3 study would be logistically and financially challenging in a setting like urban Blantyre, where health systems budgets are severely constrained [20, 22]. Mathematical modelling work in Rio de Janeiro demonstrated that reducing TB transmission within TB hotspot neighbourhoods could reduce the city-wide TB incidence [13].

Both the distance to the nearest TB clinic and the percentage of household heads who did not finish primary school were positively associated with lower case notification rates. Both the notification and prevalence rate models had random intercept standard deviations with lower bounds of their 95% credible intervals that were well away from zero, indicating that TB epidemiology varied by city neighbourhood even after adjusting for covariates. We considered modelling neighbourhood random effects with the spatial ICAR term (S1 and S2 Equations and S3 and S4 Tables), but models with random intercept terms gave a better description of the data because the spatial proximity of neighbourhoods with very different historical levels of interventions caused the ICAR model to over-smooth the data (S5 and S6 Tables). In addition, we did not have enough data to support both random effects in the same model.

Our analysis had some limitations. The prevalence survey detected a low number of prevalent cases in the city. This meant that the models for prevalence had less power for identifying predictive covariates. Other work modelling regional TB prevalence has also found most potential covariates were unable to improve predictions [33]. The prevalence survey was also just done once; it is possible that if we had a repeat prevalence survey, we could have found a different distribution of TB prevalence than what was captured in this study [40]. Some groups were also under sampled by the prevalence trial, particularly men; we accounted for this in our post-stratified sensitivity analysis, showing similar results. In addition, P:N ratios based on microbiologically-confirmed notified TB had higher mean rate ratios than in the sensitivity analysis that included all forms of TB, although the neighbourhoods identified with high P:N ratios were similar. The lower bound of the 95% CrI of some of the neighbourhood P:N ratios were less than one (Fig 3), although this represents low statistical power rather than a possibility of "overdiagnosis" of TB. More details on the goodness of fit of the models have been provided in the supplementary section (S3 Equation), which suggests that our models sufficiently describe the data well. Antiretroviral therapy (ART) coverage data was also not included in the model but there was a high percentage coverage of ART for people living with HIV across all neighbourhoods (mean: 95.41%, range: 84.01–95.41, sd: 0.03); we instead included HIV prevalence as people living with HIV are still at an increased risk of TB compared to HIV negative individuals even when they are on ART [1].

Countries in WHO Africa region need to accelerate the rate of TB incidence reduction from the current rate of about 4% per year to at least 10% by 2025 in order to meet the End TB goals [1, 2, 41]. For this to be achievable it is important that we have effective methods for prioritising communities for TB interventions to efficiently use the available resources [3]. National TB programmes that need to prioritise neighbourhood areas for TB interventions such as ACF, can collect the variables identified by our method which will be used by the model to predict the P:N ratios. By focusing on underserved communities, this will ensure universal health coverage for communities that are underserved by facility-based health care. The P:N ratios should be interpreted alongside the CNRs to obtain the full nature of the epidemic i.e., Fig 3. There is also a need to externally validate the model, as well as to investigate the effectiveness of spatially targeted interventions in randomised controlled trials [36].

## Conclusion

Using citywide enhanced surveillance data and prevalence survey data, we developed a predictive model to prioritise neighbourhoods for TB case detection and prevention activities based on readily available local data. In most low-resource settings, current active case-finding strategies are inefficient and resource-intensive. We have demonstrated a method for identifying neighbourhoods with high rates of underdiagnosis. Researchers and programme managers could prioritise identified TB hotspots for TB control and prevention interventions to focus efforts on urban TB elimination.

## Supporting information

**S1 Equation.**
(PDF)

**S2 Equation.**
(PDF)

**S3 Equation. Model goodness of fit assessment.**
(PDF)

**S1 Fig. Neighbourhood level TB prevalence to notification rate ratios (with 95% CIs) using final models.** The neighbourhoods were ordered according to prevalence to notification ratio size. Analysis based on post stratified TB prevalence with microbiologically-confirmed TB notifications kept the same as in the primary analysis. The dashed line is the mean prevalence to notification ratio. Crl Credible interval.
(PDF)

**S2 Fig. Differences of neighbourhood level TB prevalence to notification rate ratios (P:N rate ratios), the P:N rate ratio based on post stratified prevalence TB subtracted by the P:N ratio based on the primary analysis.**
(PDF)

**S3 Fig. Map of TB prevalence to notification ratios predicted from final models (Inset map of Malawi with Blantyre in red).** Analysis based on post stratified TB prevalence and with microbiologically-confirmed TB notifications kept the same as in the primary analysis. Models include neighbourhood random effects. Neighbourhoods outlined in blue are in the highest quartile for P:N ratios. Map tile data from OpenStreetMap.
(PDF)

**S4 Fig. Neighbourhood level TB prevalence to notification rate ratios (with 95% CIs) using final models.** The neighbourhoods were ordered according to prevalence to notification ratio size. Analysis based on microbiologically-confirmed TB and clinically-diagnosed cases and with TB prevalence kept the same as in the primary analysis. The dashed line is the mean prevalence to notification ratio. Crl Credible interval.
(PDF)

**S5 Fig. Map of TB prevalence to notification ratios predicted from final models (Inset map of Malawi with Blantyre in red).** Analysis based on microbiologically-confirmed TB and clinically-diagnosed cases and with TB prevalence kept the same as in the primary analysis. Models include neighbourhood random effects. Neighbourhoods outlined in blue are in the highest quartile for P:N ratios. Map tile data from OpenStreetMap.
(PDF)

**S6 Fig. Observed versus predicted mean CNRs (95% Crls).** Analysis based on microbiologically-confirmed TB as in the primary analysis. Crl Credible interval.
(PDF)

**S7 Fig. Observed versus predicted mean prevalence rates (95% Crls).** Analysis based on microbiologically-confirmed TB as in the primary analysis. Crl Credible interval.
(PDF)

**S1 Table. Table of all the TB prevalence neighbourhood level models with a random intercept of clinic of treatment registration.** Coefficients (mean rate ratio) were exponentiated

and intercepts were multiplied by 100,000.
(PDF)

**S2 Table. Table of all the TB notified neighbourhood level models with a random intercept of clinic of treatment registration.** Coefficients (mean rate ratio) were exponentiated and intercepts were multiplied by 100,000.
(PDF)

**S3 Table. Table of all the TB prevalence neighbourhood level models with spatial random effect.** Coefficients (mean rate ratio) were exponentiated and intercepts were multiplied by 100,000.
(PDF)

**S4 Table. Table of all the TB notified neighbourhood level models with spatial random effect.** Coefficients (mean rate ratio) were exponentiated and intercepts were multiplied by 100,000.
(PDF)

**S5 Table. Table of top ten prevalence models using the ELPD LOO statistic, comparing the models in S1 and S3 Tables.**
(PDF)

**S6 Table. Table of top ten notification models using the ELPD LOO statistic, comparing the models in S2 and S4 Tables.**
(PDF)

**S7 Table. Parameter estimates for final regression models for predicting neighbourhood level TB prevalence and notifications.** Analysis based on post stratified TB prevalence and with confirmed TB notifications kept the same as in the primary analysis.
(PDF)

**S8 Table. Parameter estimates for final regression models for predicting neighbourhood level TB prevalence and notifications.** Analysis based on both microbiologically-confirmed TB and clinically-diagnosed cases and with TB prevalence kept the same as in the primary analysis.
(PDF)

**S9 Table. Comparing model results of prevalence to microbiologically confirmed notification rations primary analysis, compared to analysis based on post stratified TB prevalence and based on all TB case notifications.** Only neighbourhoods that were included in the 4th quartile were included in the table.
(PDF)

## Acknowledgments

We thank the National TB Programme and Blantyre District Health Office for their support and encouragement.

## Author Contributions

**Conceptualization:** McEwen Khundi, James R. Carpenter, Elizabeth L. Corbett, Peter MacPherson.

**Data curation:** McEwen Khundi, Helena R. A. Feasey, Rebecca Nzawa Soko, Hussein Twabi, Lingstone Chiume.

**Formal analysis:** McEwen Khundi, James R. Carpenter, Marriott Nliwasa, Rachael M. Burke, Katherine C. Horton, Peter J. Dodd, Ted Cohen, Peter MacPherson.

**Funding acquisition:** Elizabeth L. Corbett.

**Investigation:** McEwen Khundi, Elizabeth L. Corbett, Rebecca Nzawa Soko, Peter MacPherson.

**Methodology:** McEwen Khundi, James R. Carpenter, Elizabeth L. Corbett, Marriott Nliwasa, Rachael M. Burke, Katherine C. Horton, Ted Cohen, Peter MacPherson.

**Project administration:** McEwen Khundi, Elizabeth L. Corbett, Helena R. A. Feasey, Rebecca Nzawa Soko, Hussein Twabi, Lingstone Chiume.

**Resources:** James R. Carpenter, Elizabeth L. Corbett, Helena R. A. Feasey.

**Software:** McEwen Khundi, Lingstone Chiume, Peter MacPherson.

**Supervision:** McEwen Khundi, James R. Carpenter, Elizabeth L. Corbett, Peter MacPherson.

**Validation:** McEwen Khundi, James R. Carpenter, Elizabeth L. Corbett, Marriott Nliwasa, Rachael M. Burke, Peter J. Dodd, Ted Cohen, Peter MacPherson.

**Visualization:** McEwen Khundi, Peter J. Dodd, Ted Cohen, Peter MacPherson.

**Writing – original draft:** McEwen Khundi.

**Writing – review & editing:** McEwen Khundi, James R. Carpenter, Elizabeth L. Corbett, Helena R. A. Feasey, Rebecca Nzawa Soko, Marriott Nliwasa, Rachael M. Burke, Katherine C. Horton, Peter J. Dodd, Ted Cohen, Peter MacPherson.

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
