## [Decision Letter · Decision Letter 0]

22 Dec 2021

PONE-D-21-35200Neighbourhood prevalence-to-notification ratios for adult bacteriologically-confirmed tuberculosis reveals hotspots of underdiagnosis in Blantyre, MalawiPLOS ONE

Dear Dr. Khundi,

Thank you for submitting your manuscript to PLOS ONE. After careful consideration, we feel that it has merit but does not fully meet PLOS ONE’s publication criteria as it currently stands. Therefore, we invite you to submit a revised version of the manuscript that addresses the points raised during the review process. Your manuscript has been thoroughly reviewed by three independent reviewers. Please carefully respond to the concerns raised by them, particularly the comments on the methods and the interpretation.

We look forward to receiving your revised manuscript.

Kind regards,

Lukas Fenner, MD, MSc

Academic Editor

PLOS ONE

Journal Requirements:

(This work was supported by two grants from Wellcome Trust (ELC grant number WT200901/Z/16/Z) and (PM grant number 200901/Z/16/Z). JRC was funded by UK Medical Research Council (MRC) programme grant MC_UU_00004/07. PJD was supported by a fellowship from the MRC (MR/P022081/1); this UK funded award was part of the EDCTP2 programme supported by the European Union. RMB was funded by Wellcome Trust (203905/16/Z). KCH was supported by the European Research Council (757699) and UK FCDO (Leaving no-one behind: transforming gendered pathways to health for TB). The funders had no role in study design, data collection and analysis, decision to publish, or preparation of the manuscript.)

(This work was supported by two grants from Wellcome Trust https://wellcome.org/ (ELC grant number WT200901/Z/16/Z) and (PM grant number 200901/Z/16/Z). JRC was funded by UK Medical Research Council  (MRC) programme grant https://www.ukri.org/councils/mrc/guidance-for-applicants/types-of-funding-we-offer/programme-grant/ MC_UU_00004/07. PJD was supported by a fellowship from the MRC (MR/P022081/1); this UK funded award was part of the EDCTP2 programme supported by the European Union. RMB was funded by Wellcome Trust (203905/16/Z). KCH was supported by the European Research Council  https://erc.europa.eu/ (757699) and UK FCDO (Leaving no-one behind: transforming gendered pathways to health for TB)  https://www.gov.uk/government/organisations/foreign-commonwealth-development-office/about/research.

The funders had no role in study design, data collection and analysis, decision to publish, or preparation of the manuscript.)

4. We note that Figures 1, 2, and 4 in your submission contain map images which may be copyrighted. All PLOS content is published under the Creative Commons Attribution License (CC BY 4.0), which means that the manuscript, images, and Supporting Information files will be freely available online, and any third party is permitted to access, download, copy, distribute, and use these materials in any way, even commercially, with proper attribution. For these reasons, we cannot publish previously copyrighted maps or satellite images created using proprietary data, such as Google software (Google Maps, Street View, and Earth). For more information, see our copyright guidelines: http://journals.plos.org/plosone/s/licenses-and-copyright.

a. You may seek permission from the original copyright holder of Figures 1,2, and 4 to publish the content specifically under the CC BY 4.0 license.  

Reviewers' comments:

Reviewer's Responses to Questions

**Comments to the Author**

1. Is the manuscript technically sound, and do the data support the conclusions?

Reviewer #1: Partly

Reviewer #2: Partly

Reviewer #3: Yes

2. Has the statistical analysis been performed appropriately and rigorously? 

Reviewer #1: Yes

Reviewer #2: Yes

Reviewer #3: Yes

3. Have the authors made all data underlying the findings in their manuscript fully available?

Reviewer #1: No

Reviewer #2: Yes

Reviewer #3: No

4. Is the manuscript presented in an intelligible fashion and written in standard English?

Reviewer #1: Yes

Reviewer #2: Yes

Reviewer #3: Yes

5. Review Comments to the Author

Reviewer #1: In this manuscript, Khundi and colleagues explore the relationship, at a neighborhood level, between TB case notification ratio, TB prevalence, and a short list of demographic and health care access variables. They highlight the important distinction between an area’s TB notification rate and its burden of undiagnosed prevalent TB, and they demonstrate that the ratio of these two quantities varies in ways that might be expected based on access to health care. The work is overall important, rigorously performed, and in my opinion a valuable addition to PLOS One. But I also think that the work would be considerably strengthened by clarifying several aspects of the methods, results, and interpretation, as follows.

Methods:

1. Aspects of methods that need to be more clearly described:

a) In the prevalence survey, what testing was done on “confirmatory specimens”, and how was a prevalent case defined?

b) Please describe how NSO 2008 and 2018 census data were used to extrapolate the neighborhood populations from 2015 to 2019. Was a single growth rate assumed for all age groups, for example? Also clearly state the assumptions made, e.g. that growth was (presumably) assumed to be the same in all neighborhoods.

c) How are the various years’ notification data accounted for in the CNR model? Table 2 suggests that all years’ data are included, but it’s not clear to me which variables in Equation 1 correspond to different years’ notifications.

d) Is the PNR based only on the estimated CNR (and prevalence) in 2019? I assume this is the case, but I don’t see it explicitly stated in the Methods.

e) The specification of the zero-inflated prevalence model (Equation 2) is unclear to me. First, what is meant by “(so W = 0 )” and “(so Z > 0 )” in line 233? ~ (1 − ) generates some zero values for Z, but it’s unclear why this implies anything about the value of W; and similarly, ~ () will require W (but not Z) to be >0. I wasn’t sure if Z and W were reversed here, or “so” was the wrong word, or there’s something else about the model specification that I’m not understanding. Also in equation 2, In line 237, there seem to be too many “=”s. After the first curly bracket, what is it that is “= 0” or “> 0”? Is there a variable or condition missing?

2. I had a couple of questions about the chosen covariates:

- Is primary school completion known to be a reliable indicator of poverty in this setting? If not, then it would be useful to see consideration of alternative proxies.

- Why would the percentage of the population who were adults be expected to predict PNR? Or is this likely to be a proxy for some other characteristic of importance, such as geographic mobility or proximity to city center?

- In general, why was this particular set of covariates chosen?

3. Minor methods comments:

- Abstract Methods and findings: Make clear that the second sentence describes the linked prevalence survey.

- Github link doesn’t yet work.

Results:

A key finding is that PNR varied by neighborhood, and in ways that were only partially explained by the measured covariates. But to really understand these results and their implications, there are several relationships that I would like to see described or quantified:

1. Can you more clearly characterize to what extent CNR was predictive of prevalence? Statistically, this might require a single model that examines the association between CNR and prevalence. It would also be useful to see a visual representation of the CNRs and the corresponding PNRs. Figure 2 may be an attempt to show this, but it’s difficult to distinguish the color scales in the small geographic areas. One possibility is to use color in figure 3 not to show quartile (which can already be determined from location on the figure) but to show either the absolute prevalence or the absolute CNR.

2. Similarly, of the variation in PNR, can you quantify how much was explained by the measured covariates and how much was modeled as neighborhood level random effects?

3. The trend in annual CNR is notable. Was there any relationship between the rate of decline in CNR (which might suggest greater access to care, or a temporary increase in notifications out of proportion to the prevalence ) and the 2019 PNR?

4. Lines 296-297, “The rates of empirical TB notifications were higher in the city centre and were lower in the city’s outskirts, whereas empirical TB prevalence rates were higher on the outskirts of the city”: Is there any way to characterize this relationship numerically, i.e., how strong and how consistent were these patterns?

5. Due to small numbers of prevalent cases, uncertainty in the prevalence is wide, and it’s not clear how this affects the main results. How confident can we be that the neighborhood level prevalence estimates improve the identification of high-prevalence neighborhoods, compared to predictions based on neighborhood demographics? Or alternatively, how densely would prevalence surveys need to sample individual neighborhoods, in order to have reasonable certainty of improving on demographic-based models?

6. In practice, if notification data plus neighborhood characteristics will be used to guide case finding, then only prior years’ notification data will be available at the time of screening site selection. If 2019 CNRs were not yet available, how would the selected model have predicted 2019 prevalence from 2015-2018 notification data?

7. It looks like the mean distance from neighborhood centroid to the nearest clinic (1.7 km) is comparable to the size of many of the neighborhoods shown in Figs 1-2 -- suggesting that the average distance may not be a good estimate of the distance to clinic for much of a neighborhood. If this is the case, I recommend (1) evaluating how well distance to clinic from a neighborhood’s center predicts the distance to clinic for each of that neighborhood’s residents, and (2) considering whether population density or distance to city center may be a more appropriate neighborhood-level covariate than distance to health facility.

More minor, some aspects of the results that should be more clearly stated:

9. Fig 2: High max values of color scales makes variation difficult to see.

10. 60.9% adults estimate: is this in 2015 or 2019? (Or if this was measured in 2015 and assumed to also apply in 2019, this should be stated more clearly in the methods as noted above.)

11. What is meant by the “midpoint of the centre of the cluster”?

12. Table 1: Why “Estimated mean HIV prevalence” and not just “HIV prevalence” or “estimated HIV prevalence”?

13. Figure 1 (and similar): Please provide a legend for reference showing distance in km, as distances to clinic are given in km and it will be difficult for most readers to convert degrees longitude/latitude to distance.

14. Consider including units (“years”) for PNR, to convey more clearly that this is a ratio divided by an annual rate. Related to this, I would not call PNR a “rate ratio.” (Although “ratio” isn’t entirely accurate either, at least it’s more standard).

Objectives, discussion, and interpretation:

1. Within the abstract, the results presented don’t support the conclusion that neighborhood prevalence surveys provide more information about hotspots than could be predicted by neighborhood socioeconomic characteristics or health center locations. Both in the abstract and throughout the paper, this is a key question that should be addressed head-on but isn’t.

2. Relatedly, throughout the manuscript, it is unclear whether the authors are offering their finding as a basis for identifying high-prevalence neighborhoods in other settings, or whether they are suggesting that every city needs to do its own neighborhood-level prevalence survey and similar modeling analysis. This is ambiguous in the introduction lines 77-81 (which actually suggests a third option that I don’t think was intended – that they developed a model which others can then use to identifying neighborhood determinants of underdiagnosed TB in Blantyre.) There’s also a seeming contradiction starting in line 400, where the authors acknowledge resource constraints that limit prevalence surveys, yet they suggest that prevalence surveys should be conducted to improve the targeting of case-finding interventions. Please clarify the intended objective, and discuss (1) whether findings demonstrate with any certainty that a prevalence survey adds information, beyond what could be predicted using notifications plus “poverty and population density”, and (2) whether any added value of the prevalence survey is sufficient to warrant the expense.

3. Please discuss the unusually rapid changes in Blantyre’s CNR between 2015 and how this might affect the results. For example, recent intensification of case finding efforts might temporarily increase notifications out of proportion to the impact on prevalence, resulting in a lower PNR. Can you comment on what was driving these notification trends in Blantyre and how it might have differed by neighborhood?

4. Please discuss the relevance of the selected covariates versus others that might have been considered in such a model.

Reviewer #2: Overall feedback:

1. An improved ability to identify detailed geographical areas where underdiagnosis of TB is occurring is critical for efficient resource allocation in Malawi and could be applied in other urban settings. This work provides a useful tool to refine estimates of undiagnosed TB prevalence at a local level.

2. I suggest the sparseness of the prevalence data could be highlighted more as part of the context to highlight the need for this approach, as detailed prevalence data would make underdiagnosis easily determined. It was not apparent until reviewing Table 1 and being able to see that just 29 adults across the 72 neighbourhoods had bacteriologically confirmed TB cases during the 2019 survey, leaving an extremely wide uncertainty range for true undiagnosed TB prevalence, relative to the hundreds of case notifications annually through regular screening.

3. It would be extremely valuable to give some context for why case notification rates have dropped from 2015 to 2019 in these neighbourhoods – based on WHO estimates and other modelling I’m familiar with this may primarily be because of drop in TB incidence (and prevalence) over this period due to scaled up TB interventions rather than a reduction in notification rates. This is especially important as the year has such a high fixed rate as part of the TB notification model in Table 2 so it seems at least naively like changes in notifications by year could be captured on the prevalence side rather than the notification side, which might change the conclusions substantially.

4. It could be discussed that a high P:N ratio alone is not necessarily a case for prioritized spending as the characteristics that define a “high underdiagnosis” neighbourhood are largely independent of TB prevalence, and rather that the estimate of undiagnosed prevalence given the estimated P:N ratio and CNR may be a much better indicator of need than the CNR often used alone.

5. Overall this adds valuable evidence to quantify the extent to which geographical access to TB services contributes to underdiagnosis of TB.

Minor comments:

Line 160 – straight line distance from the center of neighbourhoods to the nearest TB clinic is a suitable proxy for ease of access, but I would like to suggest a future refinement could be to use tools including but not limited to AccessMod to capture accessibility in more detail.

Table 1: suggest adjusting the columns a little, e.g. Mean (sd) and Range to the left and n/N to the right as the units of per 100,000 or % apply to the mean, rather than the numerator.

Suggest some minor rearrangement of the paragraphs on lines 287 – 291 and lines 298 - 301 as there is some repetition between these paragraphs.

Line 385: missing ‘on’ between solely and notification rates.

Line 388: With a comprehensive prevalence survey, underdiagnosis becomes readily apparent. I think perhaps the greatest strength of this analysis is that having identified risk factors for underdiagnosis, recent prevalence surveys (while always extremely helpful) may become less necessary.

Line 401-402: This implies that prevalence survey data is not currently used at all in prioritizing community-based interventions and may need some slight revision.

Line 460: My interpretation would be that this method is useful for identifying neighbourhoods which are likely to have higher rates of underdiagnosis based on readily available public data, rather than identifying exact neighbourhoods.

Reviewer #3: The manuscript uses Bayesian multilevel models to estimate prevalence-to-notification ratios by combining data from registered tuberculosis (TB) patients during 2015-2019 with a survey in 2019-2020.

The paper is well written and I want to commend the authors on addressing the important issue of underreporting of TB cases, but I think that there are some important issues that have to be addressed.

Major comments:

1. The link to the GitHub repository is not working anymore.

2. You use two independent models: a Poisson model for the confirmed TB case notification rates and a zero-inflated Poisson model for the prevalence data and then you calculate posterior P:N rate ratios by dividing the posterior prevalence rates by the posterior notification rates. Have you considered using a hierarchical model by fitting both models in a common framework? By calculating the P:N ratios on the posterior samples, you can still propagate your uncertainty when using two independent models, but performing the model fitting in two disjoint steps seems somewhat incoherent to me.

3. Could you explain the rationale of equation 3 in more detail? Why do you only include the fixed and random intercept parameters for both models when calculating posterior prevalence rates and posterior notification rates? In particular, it would make more sense to me to use the posterior notification rates for 2019 since your survey was performed in 2019. Isn’t there a risk that the overall decline in bacteriologically-confirmed TB cases biases your results if you only take the intercept parameters to calculate posterior notification rates? Or did you define the covariates in a way that 2019 is the reference category?

4. Did you perform posterior predictive checks to assess model fit for the two models? In MacPherson (2019, BMC Medicine), you plotted predicted and observed TB cases in Additional File 4. I think that such a comparison would also be very helpful to assess the model fit for the models you use here.

5. It seems as if you considered other covariates in MacPherson (2019, BMC Medicine) for the Poisson model for confirmed TB case notification rates (for instance percentage of population living on less than 2 US dollars) and you used a Besag-York-Mollié prior for the spatial conditional autoregressive structure in TB CNRs. Can you explain why you made these changes in the modelling of TB case notification rates?

6. Is it reasonable to assume that prevalence to notification ratios can be lower than one, i.e. can over diagnosis of TB cases occur? I am just wondering whether the lower credibility intervals for the Prevalence to notification ratios for the neighborhoods with the lowest quartile in Figure 3 make sense.

6. PLOS authors have the option to publish the peer review history of their article (what does this mean?). If published, this will include your full peer review and any attached files.

Reviewer #1: No

Reviewer #2: No

Reviewer #3: **Yes: **Sabine Hoffmann

---

## [Author Response · Author response to Decision Letter 0]

10 Mar 2022

Thank you so much for the detailed review of our work. Find attached our thorough response to the queries. We really appreciate the attention that was given to our work. Thank you.

---

## [Decision Letter · Decision Letter 1]

25 Mar 2022

PONE-D-21-35200R1Neighbourhood prevalence-to-notification ratios for adult bacteriologically-confirmed tuberculosis reveals hotspots of underdiagnosis in Blantyre, MalawiPLOS ONE

Dear Dr. Khundi,

Thank you for submitting your manuscript to PLOS ONE. After careful consideration, we feel that it has merit but does not fully meet PLOS ONE’s publication criteria as it currently stands. Therefore, we invite you to submit a revised version of the manuscript that addresses the points raised during the review process.

The reviewers acknowledge the efforts by the authors to address the comments and concerns. However, there are a few remaining issues to be addressed before it can be published.

We look forward to receiving your revised manuscript.

Kind regards,

Lukas Fenner, MD, MSc

Academic Editor

PLOS ONE

Journal Requirements:

Reviewers' comments:

Reviewer's Responses to Questions

**Comments to the Author**

1. If the authors have adequately addressed your comments raised in a previous round of review and you feel that this manuscript is now acceptable for publication, you may indicate that here to bypass the “Comments to the Author” section, enter your conflict of interest statement in the “Confidential to Editor” section, and submit your "Accept" recommendation.

Reviewer #1: (No Response)

Reviewer #2: All comments have been addressed

Reviewer #3: All comments have been addressed

2. Is the manuscript technically sound, and do the data support the conclusions?

Reviewer #1: Yes

Reviewer #2: Yes

Reviewer #3: Yes

3. Has the statistical analysis been performed appropriately and rigorously? 

Reviewer #1: Yes

Reviewer #2: Yes

Reviewer #3: Yes

4. Have the authors made all data underlying the findings in their manuscript fully available?

Reviewer #1: Yes

Reviewer #2: Yes

Reviewer #3: Yes

5. Is the manuscript presented in an intelligible fashion and written in standard English?

Reviewer #1: Yes

Reviewer #2: Yes

Reviewer #3: Yes

6. Review Comments to the Author

Reviewer #1: The authors have done a nice job of addressing my comments, with the exception of the following which warrant further attention:

1. In the introduction, it is now clearer than before that the authors are proposing to apply their modeling approach to other settings to allow prevalence and PNRs to be characterized with sparser prevalence sampling than would otherwise be required – for“Identification of neighbourhoods with high levels of under-diagnosis without requirement for prevalence surveys”. But this does not come through in the abstract, where it appears they are concluding that [prevalence] “surveys” should be conducted in every city. Please reword the abstract conclusion.

2. The color variation in Fig 2 remains difficult to see, making the figure unhelpful. Most neighborhoods are in the mid-low range, and the ones that are high (yellow) in early years are very small. Consider shifting the scale (so that there’s more color variation in the low range of 0-200 notifications/100k/year, and in the middle range of 500-1000/100k for prevalence) or using a scale with more color variation (e.g. rainbow).

3. I agree with Reviewer 2 that the exclusive emphasis on high P:N ratios is potentially misleading, and that it would be helpful to include discussion of the fact that CNRs and PNRs should ideally be considered together (rather than looking for high PNRs alone) when prioritizing areas for TB interventions such as active case finding.

Reviewer #2: (No Response)

Reviewer #3: Thank you for addressing my comments. The posterior predictions shown in Figure S6 and S7 actually show that the notification model and in particular the prevalence model fit the data rather poorly. In Figure S6 it seems as if for more than one third (26 out of the 72) of the neighbourhoods the 95% posterior predictive intervals do not include the observed case notification rate. For the prevalence model, the situation seems even worse because only 8 out of the 72 prevalence rates are included in the 95% posterior predictive intervals. This latter point is clearly linked to the zero-inflation part of the zero-inflated Poisson model, but it almost looks as if the zero-inflation hadn't worked properly because you would expect such poor predictions if you had a zero-inflated Poisson process and you fitted a regular Poisson model. Can the authors comment on this point? Correct me if I am naive, but I wouldn't trust models for which the posterior predictive distributions are so far off the observed data and I think that it would be important to acknowledge this in the manuscript and to suggest reasons for this discrepancy.

7. PLOS authors have the option to publish the peer review history of their article (what does this mean?). If published, this will include your full peer review and any attached files.

Reviewer #1: No

Reviewer #2: No

Reviewer #3: No

---

## [Author Response · Author response to Decision Letter 1]

27 Apr 2022

We have attached the Reviewer Response. Thank you.

---

## [Decision Letter · Decision Letter 2]

9 May 2022

Neighbourhood prevalence-to-notification ratios for adult bacteriologically-confirmed tuberculosis reveals hotspots of underdiagnosis in Blantyre, Malawi

PONE-D-21-35200R2

Dear Dr. Khundi,

We’re pleased to inform you that your manuscript has been judged scientifically suitable for publication and will be formally accepted for publication once it meets all outstanding technical requirements.

Kind regards,

Lukas Fenner, MD, MSc

Academic Editor

PLOS ONE

Reviewers' comments:

Reviewer's Responses to Questions

**Comments to the Author**

1. If the authors have adequately addressed your comments raised in a previous round of review and you feel that this manuscript is now acceptable for publication, you may indicate that here to bypass the “Comments to the Author” section, enter your conflict of interest statement in the “Confidential to Editor” section, and submit your "Accept" recommendation.

Reviewer #3: All comments have been addressed

2. Is the manuscript technically sound, and do the data support the conclusions?

Reviewer #3: Yes

3. Has the statistical analysis been performed appropriately and rigorously? 

Reviewer #3: Yes

4. Have the authors made all data underlying the findings in their manuscript fully available?

Reviewer #3: Yes

5. Is the manuscript presented in an intelligible fashion and written in standard English?

Reviewer #3: Yes

6. Review Comments to the Author

Reviewer #3: All comments have been addressed.

7. PLOS authors have the option to publish the peer review history of their article (what does this mean?). If published, this will include your full peer review and any attached files.

Reviewer #3: No

---

## [Editor Report · Acceptance letter]

13 May 2022

PONE-D-21-35200R2 

Neighbourhood prevalence-to-notification ratios for adult bacteriologically-confirmed tuberculosis reveals hotspots of underdiagnosis in Blantyre, Malawi 

Dear Dr. Khundi:

I'm pleased to inform you that your manuscript has been deemed suitable for publication in PLOS ONE. Congratulations! Your manuscript is now with our production department. 

Kind regards, 

on behalf of

Prof. Lukas Fenner 

Academic Editor

PLOS ONE